# Bayesian Optimization for Iterative Learning

**Vu Nguyen** *
University of Oxford
vu@robots.ox.ac.uk

**Sebastian Schulze** *
University of Oxford
sebastian.schulze@eng.ox.ac.uk

**Michael A. Osborne**
University of Oxford
mosb@robots.ox.ac.uk

## Abstract

The performance of deep (reinforcement) learning systems crucially depends on the choice of hyperparameters. Their tuning is notoriously expensive, typically requiring an iterative training process to run for numerous steps to convergence. Traditional tuning algorithms only consider the final performance of hyperparameters acquired after many expensive iterations and ignore intermediate information from earlier training steps. In this paper, we present a Bayesian optimization (BO) approach which exploits the iterative structure of learning algorithms for efficient hyperparameter tuning. We propose to learn an evaluation function compressing learning progress at any stage of the training process into a single numeric score according to both training success and stability. Our BO framework is then balancing the benefit of assessing a hyperparameter setting over additional training steps against their computation cost. We further increase model efficiency by selectively including scores from different training steps for any evaluated hyperparameter set. We demonstrate the efficiency of our algorithm by tuning hyperparameters for the training of deep reinforcement learning agents and convolutional neural networks. Our algorithm outperforms all existing baselines in identifying optimal hyperparameters in minimal time.

## 1 Introduction

Deep learning (DL) and deep reinforcement learning (DRL) have led to impressive breakthroughs in a broad range of applications such as game play [26, 36], motor control [43], and image recognition [20]. To maintain general applicability, these algorithms expose sets of hyperparameters to adapt their behavior to any particular task at hand. This flexibility comes at the price of having to tune an additional set of parameters – poor settings lead to drastic performance losses [11, 30, 37]. On top of being notoriously sensitive to these choices, deep (reinforcement) learning systems often have high training costs, in computational resources and time. For example, a single training run on the Atari Breakout game took approximately 75 hours on a GPU cluster [26]. Tuning DRL parameters is further complicated as only noisy evaluations of an agent's final performance are obtainable.

Bayesian optimization (BO) [12, 28, 35] has recently achieved considerable success in optimizing these hyperparameters. This approach casts the tuning process as a global optimization problem based on noisy evaluations of a black-box function $f$. BO constructs a surrogate model typically using a Gaussian process (GP) [31], over this unknown function. This GP surrogate is used to build an acquisition function [13, 44] which suggests the next hyperparameter to evaluate.

In modern machine learning (ML) algorithms [15], the training process is often conducted in an iterative manner. A natural example is given by deep learning where training is usually based on stochastic gradient descent and other iterative procedures. Similarly, the training of reinforcement learning agents is mostly carried out using multiple episodes. The knowledge accumulated during these training iterations can be useful to inform BO. However, most existing BO approaches [35]

---

define the objective function as the average performance over the final training iterations. In doing so, they ignore the information contained in the preceding training steps.

In this paper, we present a Bayesian optimization approach for tuning algorithms where iterative learning is available – the cases of deep learning and deep reinforcement learning. First, we consider the joint space of input hyperparameters and number of training iterations to capture the learning progress at different time steps in the training process. We then propose to transform the whole training curve into a numeric score according to user preference. To learn across the joint space efficiently, we introduce a data augmentation technique leveraging intermediate information from the iterative process. By exploiting the iterative structure of training procedures, we encourage our algorithm to consider running a larger number of cheap (but high-utility) experiments, when cost-ignorant algorithms would only be able to run a few expensive ones. We demonstrate the efficiency of our algorithm on training DRL agents on several well-known benchmarks as well as the training of convolutional neural networks. In particular, our algorithm outperforms existing baselines in finding the best hyperparameter in terms of wall-clock time. Our main contributions are:

- an algorithm to optimize the learning curve of a ML algorithm by using training curve compression, instead of averaged final performance;

- an approach to learn the compression curve from the data and a data augmentation technique for increased sample-efficiency;

- demonstration on tuning DRL and convolutional neural networks.

## 2 Related Work in Iteration-Efficient Bayesian Optimization

The first algorithm category employs stopping criteria to terminate some training runs early and allocate resources towards more promising settings. These criteria typically involve projecting towards a final score from early training stages. Freeze-thaw BO [42] models the training loss over time using a GP regressor under the assumption that the training loss roughly follows an exponential decay. Based on this projection, training resources are allocated to the most promising settings. Hyperband [8, 23] dynamically allocates computational resources (e.g. training epochs or dataset size) through random sampling and eliminates under-performing hyperparameter settings by successive halving.

Attempts have also been made to improve the epoch efficiency of other hyperparameter optimization algorithms in [5, 7, 18] which predict the final learning outcome based on partially trained learning curves to identify hyperparameter settings that are expected to under-perform and early-stop them. In the context of DRL, however, these stopping criteria, including the exponential decay assumed in Freeze-thaw BO [42], may not be applicable, due to the unpredictable fluctuations of DRL reward curves. In the supplement, we illustrate the noisiness of DRL training.

The second category [16, 17, 23, 41, 48] aims to reduce the resource consumption of BO by utilizing low-fidelity functions which can be obtained by using a subset of the training data or by training the ML model for a small number of iterations. Multi-task BO [41] requires the user to define a division of the dataset into pre-defined and discrete subtasks. Multi-fidelity BO with continuous approximation (BOCA) [16] and hierarchical partition [34] extend this idea to continuous settings. Specifically, BOCA first selects the hyperparameter input and then the corresponding fidelity to be evaluated at. The fidelity in this context refers to the use of different number of learning iterations. Analogous to BOCA's consideration of continuous fidelities, Fabolas [17] proposes to model the combined space of input hyperparameter and dataset size and then select the optimal input and dataset size jointly.

The above approaches typically identify performance of hyperparameters via the average (either training or validation) loss of the last learning iterations. Thereby, they do not account for potential noise in the learning process (e.g., they might select unstable settings that jumped to high performance in the last couple of iterations).

## 3 Bayesian Optimization for Iterative Learning (BOIL)

**Problem setting.** We consider training a machine learning algorithm given a $d$-dimensional hyperparameter $\mathbf{x} \in \mathscr{X} \subset \mathscr{R}^d$ for $t$ iterations. This process has a training time cost $c(\mathbf{x}, t)$ and produces

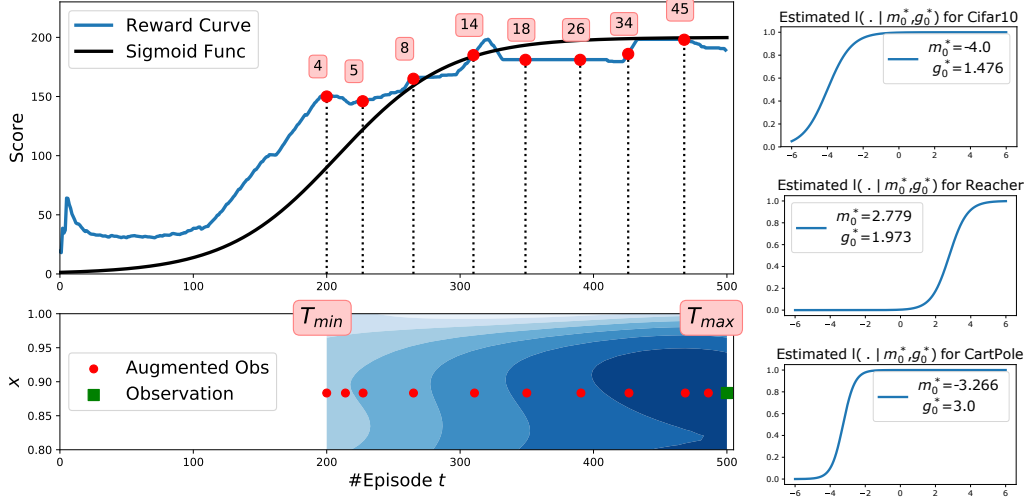

Figure 1: Left: the score in pink box is a convolution of the reward curve $r(\cdot \mid \mathbf{x} = 0.9, t = 500)$ and a Sigmoid function $l(u \mid g_0, m_0) = \frac{1}{1+\exp(-g_0[u-m_0])}$ up to time step t. Bottom: observations are selected to augment the dataset (red dots). The heatmap indicates the GP predictive mean $\mu$ for $f$ across the number of episodes $t$ used to train an agent. $T_{\min}$ and $T_{\max}$ are two user-defined thresholds for the number of training episodes. $x$ is a hyperparameter to be tuned. Right: we learn the optimal parameter $g_0^*$ and $m_0^*$ for each experiment separately.

training evaluations $r(\cdot \mid \mathbf{x}, t)$ for $t$ iterations, $t \in [T_{\min}, T_{\max}]$. These could be episode rewards in DRL or training accuracies in DL. An important property of iterative training is that we know the whole curve at preceding steps $r(t' \mid \mathbf{x}, t), \forall t' \leq t$.

Given the raw training curve $r(\cdot \mid \mathbf{x}, t)$, we assume an underlying smoothed black-box function $f$, defined in Sec. 3.2. Formally, we aim to find $\mathbf{x}^* = \arg\max_{\mathbf{x} \in X} f(\mathbf{x}, T_{\max})$; at the same time, we want to keep the overall training time, $\sum_{i=1}^N c(\mathbf{x}_i, t_i)$, of evaluated settings $[\mathbf{x}_i, t_i]$ as low as possible. We summarize our variables in Table 1 in the supplement for ease of reading.

## 3.1 Selecting a next point using iteration-efficient modeling

We follow popular designs in [17, 19, 39, 41] and model the cost-sensitive black-box function as $f(\mathbf{x}, t) \sim \mathrm{GP}\left(0, k([\mathbf{x}, t], [\mathbf{x}', t'])\right)$, where $k$ is an appropriate covariance functions and $[\mathbf{x}, t] \in \mathcal{R}^{d+1}$. For simplicity and robustness, the cost function $c(\mathbf{x}, t)$ is approximated by a linear regressor. Depending on the setting, it may be more appropriate to employ a second GP or different parametric model if the cost has a more complex dependence on hyperparameters $\mathbf{x}$ and iterations $t$. We regularly (re-)optimize both kernel and cost function parameters in between point acquisitions.

More specifically, we choose the covariance function as a product $k([\mathbf{x}, t], [\mathbf{x}', t']) = k(\mathbf{x}, \mathbf{x}') \times k(t, t')$ to induce joint similarities over parameter and iteration space. We estimate the predictive mean and uncertainty for a GP [31] at any input $\mathbf{z}_* = [\mathbf{x}_*, t_*]$ as

$$\mu(\mathbf{z}_*) = \mathbf{k}_* \left[\mathbf{K} + \sigma_y^2 \mathbf{I}\right]^{-1} \mathbf{y} \qquad (1) \qquad \sigma^2(\mathbf{z}_*) = k_{**} - \mathbf{k}_* \left[\mathbf{K} + \sigma_y^2 \mathbf{I}\right]^{-1} \mathbf{k}_*^T \qquad (2)$$

where $\mathbf{y} = [y_i]_{\forall i}$, $\mathbf{k}_* = [k(\mathbf{z}_*, \mathbf{z}_i)]_{\forall i}$, $\mathbf{K} = [k(\mathbf{z}_i, \mathbf{z}_j)]_{\forall i, j}$, $k_{**} = k(\mathbf{z}_*, \mathbf{z}_*)$, and $\sigma_y^2$ is the noise variance of $f$. Cost predictions at any particular parameter $\mathbf{x}$ and time $t$ are given by $\mu_c([\mathbf{x}_*, t_*]) = \beta^T [\mathbf{x}, t]$, where $\beta$ is directly computed from data $\{Z = [\mathbf{x}_i, t_i], \mathbf{c} = [c_i]\}_{\forall i}$ as $\beta = (Z^T Z)^{-1} Z\mathbf{c}$ [1].

Our goal is to select a point with high function value (exploitation), high uncertainty (exploration) and low cost (cheap). At each iteration $n$, we query the input parameter $\mathbf{x}_n$ and the number of iteration $t_n$ [38, 48]:

$$\mathbf{z}_n = [\mathbf{x}_n, t_n] = \underset{\mathbf{x} \in \mathcal{X}, t \in [T_{\min}, T_{\max}]}{\arg\max} \alpha(\mathbf{x}, t) / \mu_c(\mathbf{x}, t). \qquad (3)$$

Although our framework is available for any acquisition choices [13, 22, 47], to cope with output noise, we follow [45] and slight modify the expected improvement criterion using the maximum mean GP prediction $\mu_n^{\max}$. Let $\lambda = \frac{\mu_n(\mathbf{z}) - \mu_n^{\max}}{\sigma_n(\mathbf{z})}$, we then have a closed-form for the new expected improvement (EI) as $\alpha_n^{\text{EI}}(\mathbf{z}) = \sigma_n(\mathbf{z})\phi(\lambda) + [\mu_n(\mathbf{z}) - \mu_n^{\max}]\Phi(\lambda)$ where $\phi$ is the standard normal p.d.f., $\Phi$ is the c.d.f, $\mu_n$ and $\sigma_n$ are the GP predictive mean and variance defined in Eq. (1) and Eq. (2), respectively.

## 3.2 Training curve compression and estimating the transformation function

Existing BO approaches [4, 23] typically define the objective function as an average loss over the final learning episodes. However, this does not take into consideration how stable performance is or the training stage at which it has been achieved. We argue that averaging learning losses is likely misleading due to the noise and fluctuations of our observations (learning curves) – particularly during the early stages of training. We propose to compress the whole learning curve into a numeric score via a preference function representing the user's desired training curve. In the following, we use the Sigmoid function (specifically the Logistic function) to compute the utility score as

$$y = \hat{y}(r, m_0, g_0) = r(\cdot \mid \mathbf{x}, t) \bullet l(\cdot \mid m_0, g_0) = \sum_{u=1}^{t} \frac{r(u \mid \mathbf{x}, t)}{1 + \exp(-g_0[u - m_0])} \tag{4}$$

where $\bullet$ is a dot product, a Logistic function $l(\cdot \mid m_0, g_0)$ is parameterized by a growth parameter $g_0$ defining a slope and the middle point of the curve $m_0$. The optimal parameters $g_0$ and $m_0$ are estimated directly from the data. We illustrate different shapes of $l$ parameterized by $g_0$ and $m_0$ in the appendix. The Sigmoid preference has a number of desirable properties. As early weights are small, less credit is given to fluctuations at the initial stages, making it less likely for our surrogate to be biased towards randomly well performing settings. However, as weights monotonically increase, hyperparameters with improving performance are preferred. As weights saturate over time, stable, high performing configurations are preferred over short "performance spikes" characteristic of unstable training. Lastly, this utility score assigns higher values to the same performance if it is being maintained over more episodes.

**Learning the transformation function from data.** Different compression curves $l()$, parameterized by different choices of $g_0$ and $m_0$ in Eq. (4), may lead to different utilities $y$ and thus affect the performance. The optimal values of $g_0^*$ and $m_0^*$ are unknown in advance. Therefore, we propose to learn these values $g_0^*$ and $m_0^*$ directly from the data. Our intuition is that the 'optimal' compression curve $l(m_0^*, g_0^*)$ will lead to a better fit of the GP. This better GP surrogate model, thus, will result in better prediction as well as optimization performance. We parameterize the GP log marginal likelihood $L$ [31] as the function of $m_0$ and $g_0$:

$$L(m_0, g_0) = \frac{1}{2}\hat{\mathbf{y}}^T\left(K + \sigma_y^2\mathbf{I}\right)^{-1}\hat{\mathbf{y}} - \frac{1}{2}\ln|K + \sigma_y^2\mathbf{I}| + \text{const} \tag{5}$$

where $\sigma_y^2$ is the output noise variance, $\hat{\mathbf{y}}$ is the function of $m_0$ and $g_0$ defined in Eq. (4). We optimize $m_0$ and $g_0$ (jointly with other GP hyperparameters) using multi-start gradient descent. We derive the derivative $\frac{\partial L}{\partial m_0} = \frac{\partial L}{\partial \hat{y}}\frac{\partial \hat{y}}{\partial m_0}$ and $\frac{\partial L}{\partial g_0} = \frac{\partial L}{\partial \hat{y}}\frac{\partial \hat{y}}{\partial g_0}$ which can be computed analytically as:

$$\frac{\partial L}{\partial \hat{y}} = \left(K + \sigma_y^2\mathbf{I}_N\right)^{-1}\hat{y}; \quad \frac{\partial \hat{y}}{\partial m_0} = \frac{-g_0 \times \exp(-g_0[u - m_0])}{[1 + \exp(-g_0[u - m_0])]^2}; \quad \frac{\partial \hat{y}}{\partial g_0} = \frac{-m_0 \times \exp(-g_0[u - m_0])}{[1 + \exp(-g_0[u - m_0])]^2}.$$

The estimated compression curves are illustrated in Right Fig. 1 and in Sec. 4.1.

## 3.3 Augmenting the training data

When evaluating a parameter $\mathbf{x}$ over $t$ iterations, we obtain not only a final score but also all reward sequences $r(t' \mid \mathbf{x}, t), \forall t' = 1, ..., t$. The auxiliary information from the curve can be useful for BO. Therefore, we propose to augment the information from the curve into the sample set of our GP model.

A naïve approach for augmentation is to add a full curve of points $\{[\mathbf{x}, j], y_j\}_{j=1}^{t}$ where $y_j$ is computed using Eq. (4). However, this approach can be redundant and may impose serious issues in the conditioning of the GP covariance matrix. As we cluster

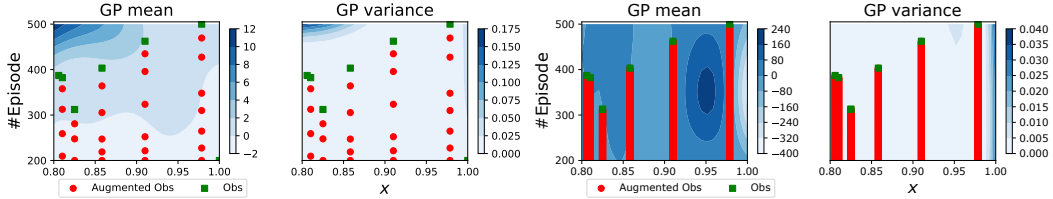

Figure 2: GP with different settings. Left: our augmentation. Right: using a full curve. If we add too many observations, the GP covariance matrix becomes ill-conditioned. On the right, the GP fit is poor with a large mean estimate range of $[-400, 240]$ even though the output is standardized $\mathcal{N}(0,1)$. All x-axis are over $x$, a hyperparameter to be tuned.

more evaluations closely, the conditioning of the GP covariance degrades further, as discussed in [24]. This conditioning issue is especially serious in our noisy DRL settings.

We highlight this effect on GP estimation in Fig. 2 wherein the GP mean varies erratically when the *natural log* of the condition number of the GP covariance matrix goes above 25 (see Fig. 3) as we include the whole curve.

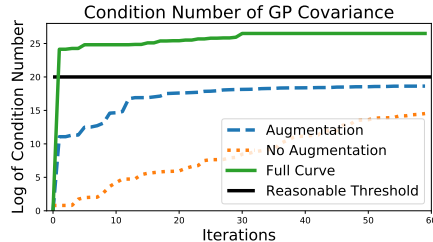

**Selecting subset of points from the curve.** Different solutions, such as the addition of artificial noise or altering the kernel's length-scales, have been proposed. We decide to use an active learning approach [10, 29] as sampled data points are expected to contain a lot of redundant information. As a consequence, the loss of information from sub-sampling the data should be minimal and information-eroding modification of the kernel matrix itself can be avoided. As a side benefit, the reduced number of sampled points speeds up inference in our GP models.

Figure 3: The condition number of GP covariance matrix deteriorates if we add the whole curve of points into a GP. The large condition number indicates the nearness to singularity.

In particular, we select samples at the maximum of the GP predictive uncertainty. Formally, we sequentially select a set $Z = [z_1, ...z_M]$, $z_m = [\mathbf{x}, t_m]$, by varying $t_m$ while keeping $\mathbf{x}$ fixed as

$$z_m = \arg\max_{\forall t' \leq t} \sigma([\mathbf{x}, t'] \mid D'), \forall m \leq M \text{ s.t. } \ln \text{ of cond}(K) \leq \delta \qquad (6)$$

where $D' = D \cup \{z_j = [\mathbf{x}, t_j]\}_{j=1}^{m-1}$. This sub-optimisation problem is done in a one-dimensional space of $t' \in \{T_{\min}, ..., t\}$, thus it is *cheap* to optimize using (multi-start) gradient descent (the derivative of GP predictive variance is available [31]). Alternatively, a fixed-size grid could be considered, but this could cause conditioning issues when a point in the grid $[\mathbf{x}, t_{\text{grid}}]$ is placed near another existing point $[\mathbf{x}', t_{\text{grid}}]$, i.e., $||\mathbf{x} - \mathbf{x}'||_2 \leq \varepsilon$ for some small $\varepsilon$.

These generated points $Z$ are used to calculate the output $r(z_m)$ and augmented into the observation set $(X, Y)$ for fitting the GP. The number of samples $M$ is adaptively chosen such that the natural log of the condition number of the covariance matrix is less than a threshold. This is to ensure that the GP covariance matrix condition number behaves well by reducing the number of unnecessary points added to the GP at later stages. We compute the utility score $y_m$ given $z_m$ for each augmented point using Eq. (4). In addition, we can estimate the running time $c_m$ using the predictive mean $\mu_c(z_m)$. We illustrate the augmented observations and estimated scores in Fig. 1.

We summarize the overall algorithm in Alg. 1. To enforce non-negativity and numerical stability, we make use of the transformations $\alpha \leftarrow \log[1 + \exp(\alpha)]$ and $\mu_c \leftarrow \log[1 + \exp(\mu_c)]$.

## 4 Experiments

We assess our model by tuning hyperparameters for two DRL agents on three environments and a CNN on two datasets. We provide additional illustrations and experiments in the appendix.

---

**Algorithm 1** Bayesian Optimization with Iterative Learning (BOIL)

---

**Input**: #iter $N$, initial data $D_0$, $\mathbf{z} = [\mathbf{x}, t]$. **Output**: optimal $\mathbf{x}^*$ and $y^* = \max_{\forall y \in D_N} y$

1: **for** $n = 1....N$ **do**
2:    Fit a GP to estimate $\mu_f()$, $\sigma_f()$ from Eqs. (1,2) and a LR for cost $\mu_c()$
3:    Select $\mathbf{z}_n = \arg\max_{\mathbf{x},t} \alpha(\mathbf{x},t)/\mu_c(\mathbf{x},t)$ and observe a curve $r$ and a cost $c$ from $f(\mathbf{z}_n)$
4:    Compressing the learning curve $r(\mathbf{z}_n)$ into numeric score using Eq. (4).
5:    Sample augmented points $\mathbf{z}_{n,m}, y_{n,m}, c_{n,m}, \forall m \leq M$ given the curve and $D_n$ in Eq. (6)
6:    Augment the data into $D_n$ and estimate Logistic curve hyperparameters $m_0$ and $g_0$.
7: **end for**

---

**Experimental setup.** All experimental results are averaged over 20 independent runs with different random seeds. Final performance is estimated by evaluating the chosen hyperparameter over the maximum number of iterations. All experiments are executed on a NVIDIA 1080 GTX GPU using the tensorflow-gpu Python package. The DRL environments are available through the OpenAI gym [3] and Mujoco [43]. Our DRL implementations are based on the open source from Open AI Baselines [6]. We release our implementation at `https://github.com/ntienvu/BOIL`.

We use square-exponential kernels for the GP in our model and estimate their parameters by maximizing the marginal likelihood [31]. We set the maximum number of augmented points to be $M = 15$ and a threshold for a natural log of GP condition number $\delta = 20$. We note that the optimization overhead is much less than the black-box function evaluation time.

**Baselines.** We compare with Hyperband [23] which demonstrated empirical successes in tuning deep learning applications in an iteration-efficient manner. We extend the discrete multi-task BO [41] to the continuous case – which can also be seen as continuous multi-fidelity BO [16, 39] as in our setting, they both consider cost-sensitivity and iteration-efficiency. We, therefore, label the two baselines as continuous multi-task/fidelity BO (CM-T/F-BO). We have ignored the minor difference in these settings, such as multi-task approaches jointly optimizes the fidelity and input while BOCA [16] first selects the input and then the fidelity.

Our focus is to demonstrate the effectiveness of optimizing the learning curve using compression and augmentation techniques. We therefore omit the comparison of various acquisition functions and kernel choices which can easily be used in our model. We also do not compare with Fabolas [17] which is designed to vary dataset sizes, not iteration numbers. We would expect the performance of Fabolas to be close to CM-T/F-BO. We are unable to compare with FreezeThaw as the code is not available. However, the curves in our setting are not exponential decays and thus ill-suited to their model (see last figure in the appendix). We have considered an ablation study in the appendix using a time kernel following the exponential decay proposed in Freeze-thaw method [42].

**Task descriptions.** We consider three DRL settings including a Dueling DQN (DDQN) [46] agent in the CartPole-v0 environment and Advantage Actor Critic (A2C) [25] agents in the InvertedPendulum-v2 and Reacher-v2 environments. In addition to the DRL applications, we tune 6 hyperparameters for training a convolutional neural network [21] on the SVHN dataset and CIFAR10. Due to space considerations, we refer to the appendix for further details.

## 4.1 Model illustration

We first illustrate the estimated compression function $l(m_0^*, g_0^*)$ in Right Fig. 1 from different experiments. These Logistic parameters $g_0^*$ and $m_0^*$ are estimated by maximizing the GP marginal likelihood and used for compressing the curve. We show that the estimated curve from CartPole tends to reach the highest performance much earlier than Reacher because CartPole is somewhat easier to train than Reacher.

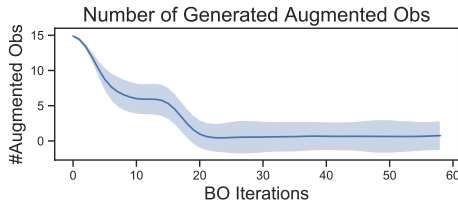

Figure 4: DDQN on CartPole. The number of augmented observations reduces over time.

We next examine the count of augmented observations generated per iteration in Fig. 4. Although this number is fluctuating, it tends to reduce over

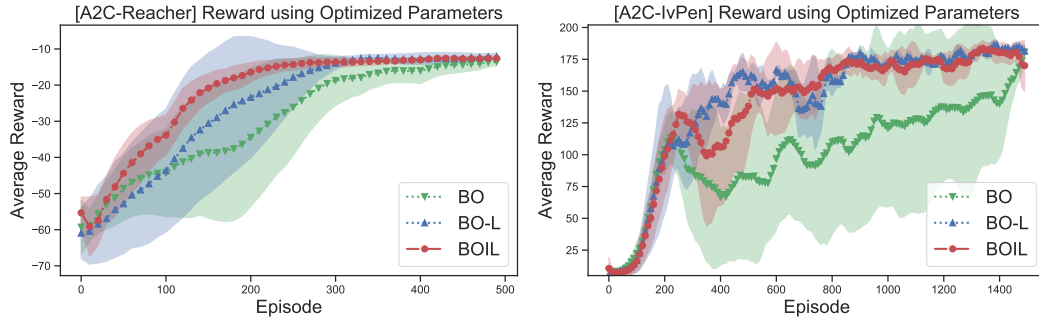

Figure 5: The learning curves of the best found parameters by different approaches. The curves show that BO-L and BOIL reliably identify parameters leading to stable training. BOIL takes only half total time to find this optimal curve.

time. BOIL does not add more augmented observations at the later stage when we have gained sufficient information and GP covariance conditioning falls below our threshold $\delta = 20$.

## 4.2 Ablation study of curve compression

To demonstrate the impact of our training curve compression, we compare BOIL to vanilla Bayesian optimization (BO) and with compression (BO-L) given the same number of iterations at $T_{\max}$. We show that using the curve compression leads to stable performance, as opposed to the existing technique of averaging the last iterations. We plot the learning curves of the best hyperparameters identified by BO, BO-L and BOIL. Fig. 5 shows the learning progress over $T_{\max}$ episodes for each of these. The curves are smoothed by averaging over 100 consecutive episodes for increased clarity. We first note that all three algorithms eventually obtain similar performance at the end of learning. However, since BO-L and BOIL take into account the preceding learning steps, they achieve higher performance more quickly. Furthermore, they achieve this more reliably as evidenced by the smaller error bars (shaded regions).

## 4.3 Tuning deep reinforcement learning and CNN

We now optimize hyperparameters for deep reinforcement learning algorithms; in fact, this application motivated the development of BOIL. The combinations of hyperparameters to be tuned, target DRL algorithm and environment can be found in the appendix.

**Comparisons by iterations and real-time.** Fig. 6 illustrates the performance of different algorithms against the number of iterations as well as real-time (the plots for CIFAR10 are in the appendix). The performance is the utility score of the best hyperparameters identified by the baselines. Across all three tasks, BOIL identifies optimal hyperparameters using significantly less computation time than other approaches.

The plots show that other approaches such as BO and BO-L can identify well-performing hyperparameters in fewer iterations than BOIL. However, they do so only considering costly, high-fidelity evaluations resulting in significantly higher evaluation times. In contrast to this behavior, BOIL accounts for the evaluation costs and chooses to initially evaluate low-fidelity settings consuming less time. This allows fast assessments of a multitude of hyperparameters. The information gathered here is then used to inform later point acquisitions. Hereby, the inclusion of augmented observations is crucial in offering useful information readily available from the data. In addition, this augmentation is essential to prevent from the GP kernel issue instead of adding the full curve of points into our GP model.

Hyperband [23] exhibits similar behavior in that it uses low fidelity (small $t$) evaluations to reduce a pool of randomly sampled configurations before evaluating at high fidelity (large $t$). To deal with noisy evaluations and other effects, this process is repeated several times. This puts Hyperband at a disadvantage particularly in the noisy DRL tasks. Since early performance fluctuates hugely, Hyperband can be misled in where to allocate evaluation effort. It is then incapable of revising

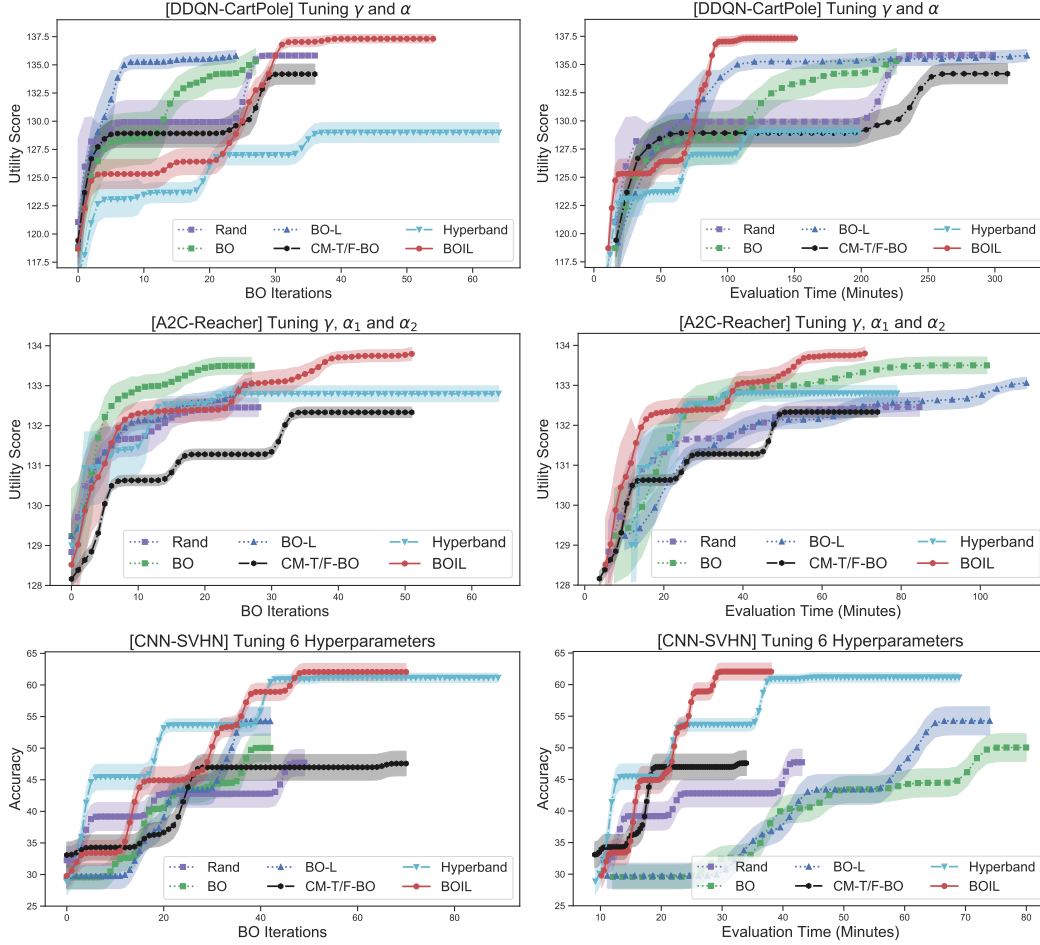

Figure 6: Comparison over BO evaluations (Left) and real-time (Right). Given the same time budget, CM-T/F-BO, Hyperband and BOIL can take more evaluations than vanilla BO, BO-L and Rand. BOIL outperforms other competitors in finding the optimal parameters in an iteration-efficient manner. CM-T/F-BO does not augment the observations from the curve and requires more evaluations. The results of InvertedPendulum and CNN-CIFAR10 are in the appendix.

these choices until an entirely new pool of hyperparameters is sampled and evaluated from scratch. In contrast to this, BOIL is more flexible than Hyperband in that it can freely explore-exploit the whole joint space. The GP surrogate hereby allows BOIL to generalize across hyperparameters and propagate information through the joint space.

# 5    Conclusion and Future work

Our framework complements the existing BO toolbox for hyperparameter tuning with iterative learning. We present a way of leveraging our understanding that later stages of the training process are informed by progress made in earlier ones. This results in a more iteration-efficient hyperparameter tuning algorithm that is applicable to a broad range of machine learning systems. We evaluate its performance on a set of diverse benchmarks. The results demonstrate that our model surpasses the performance of well-established alternatives while consuming significantly fewer resources. Finally, we note that our approach is not necessarily specific to machine learning algorithms, but more generally applies to any process exhibiting an iterative structure to be exploited.

# 6 Broader Impact

Our work aims at making the optimization of processes operating in a step-wise fashion more efficient. As demonstrated this makes BOIL particularly well-suited to supporting supervised learning models and RL systems. By increasing training efficience of these models, we hope to contribute to their widespread deployment whilst reducing the computational and therefore environmental cost their implementation has.

Deep (reinforcement) learning systems find application in a wide range of settings that directly contribute to real world decisions, e.g., natural language processing, visual task, autonomous driving and many more. As machine learning models building on our contributions are being deployed in the real world, we encourage practicioners to put in place necessary supervision and override mechanisms as precautions against potential failure.

In a more general context, our algorithm may be seen as a step towards the construction of an automated pipeline for the training and deployment of machine learning models. A potential danger is that humans become further and further removed from the modelling process, making it harder to spot (potentially critical) failures. We do not see this as an argument against the construction of such a pipeline in principle, but instead encourage practicioners to reflect on potential biases indirectly encoded in the choice of data sets and models, they are feeding into said automated processes.

The growing opacity of machine learning models is a concern of its own and which automated training procedures will only contribute to. Opposing this is a rapidly growing corpus of work addressing the interpretability of trained machine learning models and their decision making. These can and should be used to rigorously analyse final training outcomes. Only then can we ensure that machine learning algorithm do indeed become a beneficial source of information guiding real world policy making as opposed to opaque, unquestioned entities.

While our main interest lies in the hyperparameter optimization of machine learning models, it should be noted that any iterative process depending on a set of parameters can make use of our contributions. Possible settings could, for instance, include the optimization of manufacturing pipelines in which factory setting are adjusted to increase productivity.

# 7 Acknowledgements

S. Schulze is supported by an I-CASE studentship funded by the EPSRC and Dyson.

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
