[Supplementary Material]

The following sections are intended to give the reader further insights into our design choices and a deeper understanding of the algorithms properties. First, we give a brief overview of Bayesian optimization with Gaussian processes. We then illustrate our models behavior on a two dimensional problem. Last, we give further details of our experiments for reproducibility purposes.

## A Bayesian Optimization Preliminaries

Bayesian optimization is a sequential approach to global optimization of black-box functions without making use of derivatives. It uses two components: a learned surrogate model of the objective function and an acquisition function derived from the surrogate for selecting new points to inform the surrogate with. In-depth discussions beyond our brief overview can be found in recent surveys [2, 9, 35].

**Notation.** We summarize all of the notations used in our model in Table 1 for ease of reading.

### A.1 Gaussian processes

We present the GP surrogate model for the black-box function $f$ [31]. A GP defines a probability distribution over functions $f$ under the assumption that any subset of points $\{(\mathbf{x}_i, f(\mathbf{x}_i)\}$ is normally distributed. Formally, this is denoted as:

$$f(\mathbf{x}) \sim \text{GP}\left(m(\mathbf{x}), k(\mathbf{x}, \mathbf{x}')\right),$$

where $m(\mathbf{x})$ and $k(\mathbf{x}, \mathbf{x}')$ are the mean and covariance functions, given by $m(\mathbf{x}) = \mathbb{E}[f(\mathbf{x})]$ and $k(\mathbf{x}, \mathbf{x}') = \mathbb{E}\left[(f(\mathbf{x}) - m(\mathbf{x}))(f(\mathbf{x}') - m(\mathbf{x}'))^T\right]$.

Typically, the mean of the GP is assumed to be zero everywhere. The kernel $k(\mathbf{x}, \mathbf{x}')$ can be thought of as a similarity measure relating $f(\mathbf{x})$ and $f(\mathbf{x}')$. Numerous kernels encoding different prior beliefs about $f(\mathbf{x})$ have been proposed. A popular choice is given by the square exponential kernel $k(\mathbf{x}, \mathbf{x}') = \sigma_f^2 \exp\left[-(\mathbf{x} - \mathbf{x}')^2 / 2\sigma_l^2\right]$. The length- and output-scales $\sigma_l^2$ and $\sigma_f^2$ regulate the maximal covariance between two points and can be estimated using maximum marginal likelihood. The SE kernel encodes the belief that nearby points are highly correlated as it is maximized at $k(\mathbf{x}, \mathbf{x}) = \sigma_f^2$ and decays the further $\mathbf{x}$ and $\mathbf{x}'$ are separated.

For predicting $f_* = f(\mathbf{x}_*)$ at a new data point $\mathbf{x}_*$, assuming a zero mean $m(\mathbf{x}) = 0$, we have:

$$\begin{bmatrix} f \\ f_* \end{bmatrix} \sim \mathcal{N}\left(0, \begin{bmatrix} K & \mathbf{k}_*^T \\ \mathbf{k}_* & k_{**} \end{bmatrix}\right) \tag{7}$$

where $k_{**} = k(\mathbf{x}_*, \mathbf{x}_*)$, $\mathbf{k}_* = [k(\mathbf{x}_*, \mathbf{x}_i)]_{\forall i \leq N}$ and $K = [k(\mathbf{x}_i, \mathbf{x}_j)]_{\forall i, j \leq N}$. The conditional probability of $p(f_* \mid f)$ follows a univariate Gaussian distribution as $p(f_* \mid f) \sim \mathcal{N}\left(\mu(\mathbf{x}_*), \sigma^2(\mathbf{x}_*)\right)$. Its mean and variance are given by:

$$\mu(\mathbf{x}_*) = \mathbf{k}_* K^{-1} \mathbf{y}$$
$$\sigma^2(\mathbf{x}_*) = k_{**} - \mathbf{k}_* K^{-1} \mathbf{k}_*^T.$$

As GPs give full uncertainty information with any prediction, they provide a flexible nonparametric prior for Bayesian optimization. We refer the interested readers to [31] for further details on GPs.

### A.2 Acquisition function

Bayesian optimization is typically applied in settings in which the objective function is expensive to evaluate. To minimize interactions with that objective, an acquisition function is defined to reason about the selection of the next evaluation point $\mathbf{x}_{t+1} = \arg\max_{x \in \mathcal{X}} \alpha_t(\mathbf{x})$. The acquisition function is constructed from the predictive mean and variance of the surrogate to be easy to evaluate and represents the trade-off between exploration (of points with high predictive uncertainty) and exploitation (of points with high predictive mean). Thus, by design the acquisition function can be maximized with standard global optimization toolboxes. Among the many acquisition functions [12, 13, 14, 32, 40, 44] available in the literature, the expected improvement [14, 27, 45] is one of the most popular.

Table 1: Notation List

| Parameter | Domain | Meaning |
|---|---|---|
| $d$ | integer, $\mathcal{N}$ | dimension, no. of hyperparameters to be optimized |
| $\mathbf{x}$ | vector, $\mathcal{R}^d$ | input hyperparameter |
| $N$ | integer, $\mathcal{N}$ | maximum number of BO iterations |
| $T_{\min}, T_{\max}$ | integer, $\mathcal{N}$ | the min/max no of iterations for training a ML algorithm |
| $t$ | $\in [T_{\min}, ... T_{\max}]$ | index of training steps |
| $M$ | integer, $\mathcal{N}$ | the maximum number of augmentation. We set $M = 15$. |
| $\delta$ | scalar, $\mathcal{R}$ | threshold for rejecting augmentation when ln of cond$(K) > \delta$ |
| $m$ | $\in \{1, ... M\}$ | index of augmenting variables |
| $n$ | $\in \{1, ..., N\}$ | index of BO iterations |
| $\mathbf{z} = [\mathbf{x}, t]$ | vector, $\mathcal{R}^{d+1}$ | concatenation of the parameter $\mathbf{x}$ and iteration $t$ |
| $c_{n,m}$ | scalar, $\mathcal{R}$ | training cost (sec) |
| $y_n$ | scalar, $\mathcal{R}$ | transformed score at the BO iteration $n$ |
| $y_{n,m}$ | scalar, $\mathcal{R}$ | transformed score at the BO iteration $n$, training step $m$ |
| $\alpha(\mathbf{x}, t)$ | function | acquisition function for performance |
| $\mu_c(\mathbf{x}, t)$ | function | estimation of the cost by LR given $\mathbf{x}$ and $t$ |
| $r(. \mid \mathbf{x}, t)$ | function | a raw learning curve, $r(\mathbf{x}, t) = [r(1 \mid \mathbf{x}, t), ... r(t' \mid \mathbf{x}, t), r(t \mid \mathbf{x}, t)]$ |
| $f(\mathbf{x}, t)$ | function | a black-box function which is compressed from the above $f()$ |
| $l(. \mid m_0, g_0)$ | function | Logistic curve $l(u \mid m_0, g_0) = \frac{1}{1+\exp(-g_0[u-m_0])}$ |
| $g_0, g_0^*$ | scalar, $\mathcal{R}$ | a growth parameter defining a slope, $g_0^* = \arg\max_{g_0} L$ |
| $m_0, m_0^*$ | scalar, $\mathcal{R}$ | a middle point parameter, $m_0^* = \arg\max_{m_0} L$ |
| $L$ | scalar, $\mathcal{R}$ | Gaussian process log marginal likelihood |

### A.3 GP kernels and treatment of GP hyperparameters

We present the GP kernels and treatment of GP hyperparameters for the black-box function $f$.

Although the raw learning curve in DRL is noisy, the transformed version using our proposed curve compression is smooth. Therefore, we use two squared exponential kernels for input hyperparameter and training iteration, respectively. That is $k_x(\mathbf{x}, \mathbf{x}') = \exp\left(-\frac{||\mathbf{x}-\mathbf{x}'||^2}{2\sigma_x^2}\right)$ and $k_t(t, t') = \exp\left(-\frac{||t-t'||^2}{2\sigma_t^2}\right)$ where the observation $\mathbf{x}$ and $t$ are normalized to $[0, 1]^d$ and the outcome $y$ is standardized $y \sim \mathcal{N}(0, 1)$ for robustness. As a result, our product kernel becomes

$$k\left([\mathbf{x}, t], [\mathbf{x}', t']\right) = k(\mathbf{x}, \mathbf{x}') \times k(t, t') = \exp\left(-\frac{||\mathbf{x}-\mathbf{x}'||^2}{2\sigma_x^2} - \frac{||t-t'||^2}{2\sigma_t^2}\right).$$

The length-scales $\sigma_x$ and $\sigma_t$ are learnable parameters indicating the variability of the function with regards to the hyperparameter input $\mathbf{x}$ and number of training iterations $t$. Estimating appropriate values for them is critical as this represents the GPs prior regarding the sensitivity of performance w.r.t. changes in the number of training iterations and hyperparameters. For extremely large $\sigma_t$ we expect the objective function to change very little for different numbers of training iterations. For small $\sigma_t$ by contrast we expect drastic changes even for small differences. We estimate these GP hyperparameters (including the length-scales $\sigma_x$, $\sigma_t$ and the output noise variance $\sigma_y$) by maximizing their log marginal likelihood [31].

We optimize Eq. (5) with a gradient-based optimizer, providing the analytical gradient to the algorithm. We start the optimization from the previous hyperparameter values $\theta_{prev}$. If the optimization fails due to numerical issues, we keep the previous value of the hyperparameters. We refit the hyperparameters every $3 \times d$ function evaluations where $d$ is the dimension.

## B   Algorithm Illustration and Further Experiments

Fig. 7 and Fig. 8 illustrate the behavior of our proposed algorithm BOIL on the example of optimizing the discount factor $\gamma$ of Dueling DQN [46] on the CartPole problem. The two settings differ in the inclusion augmented observations into BOIL in Fig. 7 and CM-T/F-BO (or BOIL without augmented observations) in Fig. 8.

Figure 7: Illustration of BOIL on a 2-dimensional optimization task of DDQN on CartPole. The augmented observations fill the joint hyperparameter-iteration space quickly to inform our surrogate. Our decision balances utility $\alpha$ against cost $\tau$ for iteration-efficiency. Especially in situations of multiple locations sharing the same utility value, our algorithm prefers to select the cheapest option.

Table 2: Dueling DQN algorithm on CartPole problem.

| Variables | Min | Max | Best Found $\mathbf{x}^*$ |
|---|---|---|---|
| $\gamma$ discount factor | 0.8 | 1 | 0.95586 |
| learning rate model | $1e^{-6}$ | 0.01 | 0.00589 |
| #Episodes | 300 | 800 | - |

Figure 8: Illustration of the Continuous Multi task/fidelity BO (CM-T/F-BO) -- this is the case of BOIL **without** using augmented observations (same setting as Fig. 7). This version leads to less efficient optimization as the additional iteration dimension requires more evaluation than optimizing the hyperparameters on their own.

Table 3: A2C algorithm on Reacher (left) and InvertedPendulum (right).

| Variables | Min | Max | Best Found $\mathbf{x}^*$ | Min | Max | Best Found $\mathbf{x}^*$ |
|---|---|---|---|---|---|---|
| $\gamma$ discount factor | 0.8 | 1 | 0.8 | 0.8 | 1 | 0.95586 |
| learning rate actor | $1e^{-6}$ | 0.01 | 0.00071 | $1e^{-6}$ | 0.01 | 0.00589 |
| learning rate critic | $1e^{-6}$ | 0.01 | 0.00042 | $1e^{-6}$ | 0.01 | 0.00037 |
| #Episodes | 200 | 500 | - | 700 | 1500 | - |

Table 4: Convolutional Neural Network.

| Variables | Min | Max | Best Found $\mathbf{x}^*$ |
|---|---|---|---|
| filter size | 1 | 8 | 5 |
| pool size | 1 | 5 | 5 |
| batch size | 16 | 1000 | 8 |
| learning rate | $1e^{-6}$ | 0.01 | 0.000484 |
| momentum | 0.8 | 0.999 | 0.82852 |
| decay | 0.9 | 0.999 | 0.9746 |
| number of epoch | 30 | 150 | - |

In both cases, we plot the GP predictive mean in Eq. (1), GP predictive variance in Eq. (2), the acquisition function in Eq. (3), the predicted function and the final decision function in Eq. (8). These equations are defined in the main manuscript.

As shown in the respective figures the final decision function balances between utility and cost of any pair $(\gamma, t)$ to achieve iteration efficiency. Especially in situations where multiple locations share the same utility value, our decision will prefer to select the cheapest option. Using the augmented observations in Fig. 7, our joint space is filled quicker with points and the uncertainty (GP variance) across it reduces faster than in Fig. 8 – the case of vanilla CM-T/F-BO without augmenting observations. A second advantage of having augmented observations is that the algorithm is discouraged to select the same hyperparameter setting at lower fidelity than a previous evaluation. We do not add the full curve as it can be redundant while causing the conditioning problem of the GP covariance matrix.

## B.1 Experiment settings

We summarize the hyperparameter search ranges for A2C on Reacher and InvertedPendulum in Table 3, CNN on SHVN in Table 4 and DDQN on CartPole in Table 2. Additionally, we present the best found parameter $\mathbf{x}^*$ for these problems. Further details of the DRL agents are listed in Table 5.

## B.2 Learning Logistic Function

We first present the Logistic curve $l(u \mid \mathbf{x}, t) = \frac{1}{1+\exp(-g_0[u-m_0])}$ using different choices of $g_0$ and $m_0$ in Fig. 10. We then learn from the data to get the optimal choices $g_0^*$ and $m_0^*$ presented in Fig. 11.

Table 5: Further specification for DRL agents

| Hyperparameter | Value | Dueling DQN | |
|---|---|---|---|
| | | Q-network architecture | $[50, 50]$ |
| **A2C** | | $\varepsilon$-greedy (start, final, number of steps) | $(1.0, 0.05, 10000)$ |
| Critic-network architecture | $[32, 32]$ | Buffer size | 10000 |
| Actor-network architecture | $[32, 32]$ | Batch size | 64 |
| Entropy coefficient | 0.01 | PER-$\alpha$ [33] | 1.0 |
| | | PER-$\beta$ (start, final, number of steps) | $(1.0, 0.6, 1000)$ |

Figure 9: To highlight the robustness, we examine the results using different preference functions such as Sigmoid curve, Log curve, and Average curve on Reacher experiments. The results include the best found reward curve with different preference choices that show the robustness of our model. Left column: the best found curve using averaged reward over 100 consecutive episodes. Right column: the best found curve using the original reward.

Figure 10: Examples of Logistic function $l(u) = \frac{1}{1+\exp(-g_0[u-m_0])}$ with different values of middle parameter $m_0$ and growth parameter $g_0$.

### B.3 Robustness over Different Preference Functions

We next study the learning effects with respect to different choices of the preference functions. We pick three preference functions including the Sigmoid, Log and Average to compute the utility score for each learning curve. Then, we report the best found reward curve under such choices. The experiments are tested using A2C on Reacher-v2. The results presented in Fig. 9 demonstrate the robustness of our model with the preference functions.

### B.4 Applying Freeze-Thaw BO in the settings considered

While both the exponential decay in Freeze-Thaw BO [42] and our compression function encode preferences regarding training development, there is an important distinction between the two approaches. Freeze-thaw BO utilises the exponential decay property to *terminate* the training curve, while BOIL only uses the sigmoid curve to *guide* the search. We refer to Fig. 13 for further illustration of why Freeze-thaw BO struggles in DRL settings.

### B.5 Ablation Study using Freeze-Thraw Kernel for Time

In the joint modeling framework of hyperparameter and time (iteration), we can replace the kernel either $k(\mathbf{x},\mathbf{x})$ or $k(t,t)$ with different choices. We, therefore, set up a new baseline of using the time-kernel $k(t,t')$ in Freeze-Thaw approach [42] which encodes the monotonously exponential decay from the curve. Particularly, we use the kernel defined as

$$k(t,t') = \frac{\beta^\alpha}{(t+t'+\beta)^\alpha}$$

for parameters $\alpha, \beta > 0$ which are optimized in the GP models.

Figure 11: We learn the suitable transformation curve directly from the data. We parameterized the Logistic curve as $l(m_0, g_0) = \frac{1}{1+\exp(-g_0[1-m_0])}$ then estimate $g_0$ and $m_0$. The estimated function $l(m_0^*, g_0^*)$ is then used to compress our curve. The above plots are the estimated $l()$ at different environments and datasets.

Figure 12: Tuning hyperparameters of a DRL on InvertedPendulum and a CNN model on CIFAR10.

Figure 13: Illustration of Freeze-thaw BO in DRL. Freeze-thaw BO will terminate training processes when training performance (in blue) significantly drops (i.e. at the red locations) as the *exponential decay* model will predict low final performance. In most RL enviroments noisy training curves are unavoidable. Thus, Freeze-thaw BO will dismiss all curves including good setting, never completing a single training run before the final epoch.

Figure 14: Comparison using freezethaw kernel for time component.

We present the result in Fig. 14 that CM-T/F-BO is still less competitive to BOIL using this specific time kernel. The results again validate the robustness our approach cross different choices of kernel.

## B.6 Additional Experiments for Tuning DRL and CNN

We present the additional experiments for tuning a DRL model using InvertedPendulum environment and a CNN model using a subset of CIFAR10 in Fig. 12. Again, we show that the proposed model clearly gain advantages against the baselines in tuning hyperparameters for model with iterative learning information available.

## B.7 Examples of Deep Reinforcement Learning Training Curves

Finally, we present examples of training curves produced by the deep reinforcement learning algorithm A2C in Fig. 15. These fluctuate widely and it may not be trivial to define good stopping criteria as done for other applications in previous work [42].

Figure 15: Examples of reward curves using A2C on Reacher-v2 (rows $1 - 3$) and on InvertedPendulum-v2 (rows $4 - 6$). Y-axis is the reward averaged over 100 consecutive episodes. X-axis is the episode. The noisy performance illustrated is typical of DRL settings and complicates the design of early stopping criteria. Due to the property of DRL, it is not trivial to decide when to stop the training curve. In addition, it will be misleading if we only take average over the last 100 iterations.