[Reviews · NeurIPS 2020]

Review 1

Summary and Contributions: This paper aims to increase the efficiency of BO on DL/DRL, which are both expensive and challenging to tune. The main contributions of this paper are a comparison on different DRL tuning tasks and utilizing and also learning a training curve compression.

Strengths: I find the paper interesting and practical. The experimental section is convincing and well-executed and the steps of the method are clearly explained and seem different from previous work. Fig. 5 is does a good job convincing the reader that the final parameters are more stable.

Weaknesses: Ill-conditioning: Section 3.2 seems to provides a non-intuitive solution to the underlying problem. Selecting a subset of the points through active learning seems unnecessary and it would be better to understand what causes the conditioning problems. Have you experimented with different kernels and how does this affect the issue? Baselines: I would still like to see a comparison to FABOLAS, Freeze-Thaw, MISO-KG, and BOHB to further strengthen the experimental evaluation.

Correctness: Yes

Clarity: The paper is well-written and easy to read.

Relation to Prior Work: Yes

Reproducibility: Yes

Additional Feedback: === Post-rebuttal update === Thank you for your response. The ill-conditioning likely comes from points being close together, but the decision to use one of the smoothest kernels (SE) available is questionable in this case. It would be interesting to explore a less smooth kernel in combination with removing points that are close to each other. While I still think that the active learning approach is an unnecessarily complicated solution to the problem, it seems to work. I agree with the other reviewers that this paper is borderline and offers interesting empirical results while lacking novelty and theory.


Review 2

Summary and Contributions: This paper proposes a BO method to efficiently learn the hyperparameters of deep RL systems. Their BO algorithm uses a new objective function that together with a sample selection helps improve the learning process.

Strengths: - the paper is well-written and well-structured. the abstract and introduction read very well. - I like the idea, it is novel and it has been well investigated through experiments - the paper does a great job referring to previous work of art and comparing to them

Weaknesses: I am not convinced from the experiments that the method is significantly better than other learning methods. For example, in figure 6, the performance of BOIL is only marginally better than other methods. The authors mention BOIL is much faster but then the evaluation time seem to also only offer marginal improvement.

Correctness: yes

Clarity: yes, very well-written all throughout

Relation to Prior Work: yes, the author do a great comprehensive job of comparing to other methods.

Reproducibility: Yes

Additional Feedback: Thank you for the response. That clears things up.


Review 3

Summary and Contributions: This paper presents a method for improving deep reinforcement learning hyperparameter tuning by: building a surrogate model that incorporates iteration number (similarly to a fidelity variable), predicting the query cost, compress the reward curve in a single value, augment the observations by a curve fit.

Strengths: The experimental evaluation is very correct. The experiments are interesting enough while remaining tracktable and reproducible. There is an ablation study. The state-of-the-art is properly compared.

Weaknesses: Despite the conclusions by the authors, it seems that this approach is very limited to DRL problems. For example, the authors claim that they cannot use Freeze-Thaw or Fabolas because they assume exponential decay, but in this work, a logistic function is assumed instead. Therefore, it might be applicable only if the logistic approximation is a good compression.

Correctness: This is mostly a practical paper and the empirical methodology is one of the strong points.

Clarity: The paper is hard to follow at some points. For example, Section 3.2 and 3.3 should be reversed as Section 3.2 makes continuous reference to eq 7, and even in the algorithm, the order is the opposite.

Relation to Prior Work: The related work section is one of the most clear part of the text. All the related work is properly cited, discussed and compared if needed. If the comparison is not possible, it is justified.

Reproducibility: Yes

Additional Feedback: Update: After reading the authors response and the reviews, I have updated my score. I think the authors should add Fig 1 of their response to the supplementary material, as it makes quite clear and intuitive the design and purpose of the method. However, the authors response has increased my concerns of the method being truly useful for DRL applications. Although this is not a drawback "per se", I think the manuscript would be much stronger if it embraced that "limitation" and presented it as a integrated DRL strategy. ----------------------------------- -The comment on ill-conditioning when points get closer, that is well known in the BO community since early 2000s and that is one of the reasons to avoid the squared-exponential kernel. See for example: M.J. Sasena. Flexibility and Efficiency Enhancement for Constrained Global Design Optimization with Kriging Approximations. PhD thesis, University of Michigan, 2002. -How is the reward curve considered in the CNN case? Was it also compressed using a logistic curve? Given that the initial noise in that case should be smaller than in the DRL cases, it is surprising that BO perform worse while it does ok in the DRL. -The authors cite both [19] and [6] as Hyperband, but I assume they only use the random case [19] in the comparison. Given the competitiveness of Hyperband in all cases (and when it fails, BO does excellent), it seems that the combination of both as in [6] would be a tough competitor.


Review 4

Summary and Contributions: The paper presents a practical approach that tries to circunvemnt the computational cost of hyperparametrizing learning algorithms with an iterative structure by transforming each training curve into numeric scores representing training success as well as stability and then selectively augment the data using the information from the curve. They provide a good experimental framework that provides enough empirical claims to support their arguments with examples on deep reinforcement learning and convolutional neural networks. The paper resembles freeze-thaw bayesian optimization, but I prefer this method. I think that the paper can be presented in the NeurIPS conference, given that the observations are satisfied.

Strengths: Significance Empirical Work Reproducibility (If code given)

Weaknesses: Theoretical work Novelty (Freeze Thaw Bayes Opt.)

Correctness: Yes, I think

Clarity: Yes, it is.

Relation to Prior Work: Yes

Reproducibility: Yes

Additional Feedback: Author rebuttal: I have read the author's response and talk with the other reviewers about this paper. After that, I keep my score as my most important demands were not met and reviewers conclude in giving an average of 6 for this paper. ========= -> I would have like a bit of theoretical work on the paper, are not they any simple bounds that you can stablish with respect to vanilla BO? -> I also miss a computational complexity analysis of the BOIL algorithm. -> I expect that the authors could provide the source code as it is stated in the paper in the review process. -> -> Reinforcement learning literature in the introduction is too wide and too informal. I would formalize it a bit for BO researchers that do not have experience in Reinforcement learning and make it less "narrative". The algorithm then would make more sense. You could for example give a formal representation of the iterative learning algorithms where BOIL are effective, described in an abstract way, making DRL as a concrete example of this algorithms. It would give quality to the paper -> I was familiar with Freeze Thaw Bayes. Opt. but have not considered that it was not optimal for this case. Could you provide a simple maybe synthetic or toy experiment where this can be shown? It would give a lot of points to the paper since Freeze Thaw is so popular and augment its credibility.

[Author Response · NeurIPS 2020]

We are glad that our reviewers agree on the merits and relevance of our work. We are grateful for the constructive
comments and will incorporate them into the final version of our submission. We would also like to reassure our
reviewers, that we will make an implementation of our work available upon publication.

**R3/R4: Applying Freeze-Thaw BO in the settings considered.** While both the exponential decay in Freeze-Thaw (FT)
and our compression function encode preferences regarding training development, there is an important distinction
between the two approaches. FT utilises the exponential decay property to *terminate* the training curve, while BOIL
only uses the sigmoid curve to *guide* the search. See Fig. 1 for further illustration of why FT struggles in DRL settings.

Reward Curves Examples using A2C on Inverted Pendulum

Figure 1: Illustration of Freeze-thaw (FT) in DRL. FT will terminate training processes when training performance (in blue) significantly drops (i.e. at the red locations) as the *exponential decay* model will predict low final performance. In most RL enviroments noisy training curves are unavoidable. Thus, FT will dismiss all curves including good setting, never completing a single training run before the final epoch.

**R3: Comparison to Fabolas.** Fabolas uses a different way of obtaining low-fidelity information. In particular, it
uses subsets of the training data to estimate performance on a full data set. This is *orthogonal* to our approach—a
combination of the two might make for interesting future work. In the context of DRL, however, it is unclear how it
would apply, as no fixed set of training data exists. The CM-T/F-BO work which we compare against can be seen as an
adaptation of Fabolas to run-time fidelity.

**R3: Sec 3.2 and 3.3 should be reversed as Sec 3.2 makes reference to Eq (7).**
We will revise the presentation to improve the cohesion across sections in the final version.

**R3: Different versions of Hyperband.** We have used the original implementation of Hyperband [Li et al 2018]. We
can consider the variant of [Falkner et al 2018]. However, we would highlight that (1) our curve compression idea is
unique and (2) we can extend their approaches using our compression strategy in future work.

**R1/R3: Regarding the issue of ill-conditioning and its connection to kernel choice** The ill-conditioning problem can
occur whenever sampled points give near-identical rows in the kernel Gram matrix. For stationary kernels, this is
usually the case for tightly-clustered data points. While the SE kernel does pose particular conditioning problems, we
found that the problem persisted across all popular kernel choices, including the Matérn class. Different solutions, such
as the addition of artificial noise or altering the kernel's lengthscales, have been proposed. The active learning approach
was chosen as sampled data points are expected to contain a lot of redundant information. As a consequence, the loss of
information from sub-sampling the data should be minimal and information-eroding modification of the kernel matrix
itself can be avoided. As an added benefit, the reduced number of sampled points speeds up inference in our GP models.

**R3: How is the reward curve considered in the CNN case? Was it also compressed using a logistic curve?**
Training performance in the CNN experiments is measured as the predictive performance on test data after a number of
SGD training steps. The resulting performances curves are compressed using a logistic curve whose parameters are
estimated as described and shown in Fig. 5 of the supplement.

**R2: In Fig 6, the performance of BOIL is marginally better than other methods.**
While other algorithms achieve similar final performance to BOIL, they typically require more time to do so and are
less robust across varying problem settings as shown in the right column of Fig 6. (Note that the left column shows
performance over training iterations - not real time.)

**R4: Computational complexity of the BOIL algorithm.** In the settings considered, the overhead of BOIL's computation
is significantly outweighted by the cost of executing the actual learning algorithm. For instance, a full training run on
the CartPole environment takes about 600 secs, whereas the matching BOIL iteration required about 1 sec of real time.

The highest computational cost of BOIL is incurred by inference for the GP models. This cost scales cubically in the
number of observed data points. However, due to the high cost of collecting data, we would not expect this to become a
problem. Alternatively, sparse GP approximations can alleviate such problems and scale to millions of data points.

[Meta-Review · NeurIPS 2020]

The paper proposes an idea for tuning hyper-parameters in deep (reinforcement) learning using Bayesian optimization. The key idea is to exploit the iterative structure of the problem and use a variable-augmentation trick to learn a score function that compresses the learning progress at any stage. The score accounts for training success and stability. The strengths of the paper are: - well written - good relation to prior work - good experimental study However, the paper also has weaknesses, which are mostly related to theoretical aspects and chosen heuristics (see some details below). 1. If we are only interested in the predictive mean for the cost-GP, why do we use a GP in the first place, and not parametric function, which scales much better? 2. All reviewers agree that everything related to the condition number is heuristic and unclear. That's the one part that caused us the most toothache. We don't think this part is overly critical, and that the other ideas are quite valuable. Here are our concerns: In the first place, condition numbers are unintuitive, and it depends on what you want to do with the matrix, the implementation, the floating point representation etc. in order to make (heuristic) statements about numerical stability. This is nowhere explained, and neither of us reviewers knows how to set these thresholds. The GP covariance's condition number shouldn't be too bad if the (measurement) noise is reasonably big (more precisely, the signal-to-noise ratio should not be horrible). Also, the smallest eigenvalue of K + \sigma^2 I should be attained already with 2 identical inputs. If observations are noise-free, things can go crazy w.r.t. condition numbers, obviously. In this case, a standard approach is to add a jitter term (nugget) to the kernel matrix, which would play the role of the measurement noise. Again, that's a good way to control condition numbers. A possible solution would be to just add a nugget/jitter term by default and remove the part on the condition number from the paper. 3. Optimization via gradient descent: How do you deal with multi-modality? 4. When the parameters of the score function are estimated, this is basically a (regularized) least-squares fit (using the data-fit term of the log-marginal likelihood). It would be interesting to see what effect this has on the GP model itself in terms of fitting, i.e., how do the GP hyper-parameters change compared to a vanilla GP fit. Does the GP model still make sense, i.e., is the fit good. Some insights here would be useful. Overall, we (the reviewers) think that this paper is borderline-borderline. To be honest, it's not a great paper, but there's nothing fundamentally wrong with it, and it has some interesting ideas. In the end, the question is whether a resubmission of the work will make the work significantly better or whether this just adds pain to the authors' lives without significant benefit to the paper. We don't think this paper will ever make it out of the "noise zone" (borderline zone) at conferences. In the end, we recommend to accept this paper, but we *strongly* urge the authors to consider the issues raised above (also look at the reviews) in the final version. In particular (but not the only issue): We require the authors to address the comments/feedback related to the conditioning issues. These issues were already pointed out by the reviewers for UAI 2020, but the authors didn't address these comments in their resubmission to NeurIPS. So, this is the last chance to address this issue.